# Antioxidant cysteine and methionine derivatives show trachea disruption in insects

**Hiroyuki Morimura[1], Kota Ishigami[1,2], Shusei Kanie[1], Yuya Sato[3], Yoshitomo Kikuchi[1,2]\***

**1** Bioproduction Research Institute, National Institute of Advanced Industrial Science and Technology (AIST), Hokkaido Center, Sapporo, Japan, **2** Graduate School of Agriculture, Hokkaido University, Sapporo, Japan, **3** Environmental Management Research Institute, National Institute of Advanced Industrial Science and Technology (AIST), Tsukuba Center, Tsukuba, Japan

\* y-kikuchi@aist.go.jp

## Abstract

To prevent the deterioration of the global environment, the reduction of chemical pesticide use and the development of eco-friendly pest control technologies are urgent issues. Our recent study revealed that the production of reactive oxygen species (ROS) by dual oxidase (Duox) plays a pivotal role in stabilizing the tracheal network by intermediating the tyrosine cross-linking of proteins that constitute trachea. Notably, the formation of dityrosine bonds by ROS can be inhibited by the intake of an antioxidant cysteine derivative *N*-acetyl-L-cysteine (NAC), which can suppress insect respiration. In this study, we screened for the derivatives showing insecticidal activity and tracheal formation inhibition. As a result of investigating the soybean pest bug *Riptortus pedestris*, cysteine and methionine derivatives showed respiratory formation inhibition and high insecticidal activity. In particular, NAC had a slow-acting insecticidal effect, while L-cysteine methyl ester (L-CME) showed relatively fast-acting insecticidal activity. Furthermore, the insecticidal activity of these derivatives was also detected in *Drosophila*, mealworms, cockroaches, termites, and plant bugs. Our results suggest that some antioxidant compounds have specific tracheal inhibitory activity in different insect species and they may be used as novel pest control agents upon further characterization.

## Introduction

The threat of environmental issues including global warming and chemical environmental pollution by industries, are increasing rapidly [1–7]. Concerns have been raised that chemical pesticides kill non-target animals and influence the endocrine systems of organisms as environmental hormones, affecting ecological dynamics and leading to the collapse of ecosystems [8–14]. In response to these concerns, the development of environmentally friendly pest management technologies has become a pressing matter, and the implementation of new regulations about their usage is leading to a decreasing trend in chemical insecticide usage worldwide [15–21]. Insecticide-independent pest control methods such as microbial pathogens, parasitoid wasps, predators, pheromones and light traps, and ultrasonic transducers are

**Data Availability Statement:** The datasets used and/or analyzed during the current study are available from Dryad (https://doi.org/10.5061/dryad.2280gb60h).

**Funding:** This study is financed by Japan Society for the Promotion of Science (JSPS) KAKENHI project 21K18241 and Northern Advancement Center for Science & Technology (NOASTEC) projects S-3-7 and H-3-5.The funders played no role in the study design, data collection and analysis, decision to publish, or preparation of the manuscript.

**Competing interests:** NO authors have competing interests.

promising solutions that can reduce the risk of insecticides on the agricultural ecosystem [22–29]. Although these methods may not be as fast-acting as chemical insecticides, they can provide effective and sustainable pest control [30].

To control pest insects effectively and specifically, a deep understanding of the unique features of pest insects' behavior, communication, structure, physiology, and biomolecules is pivotal [31, 32]. In terms of the unique characteristics of insects, their respiratory system differs significantly from that of vertebrates in both structure and function [33, 34]. While terrestrial vertebrates rely on their lungs to breathe, insects, and some terrestrial arthropods, exchange oxygen and carbon dioxide directly through tracheae which extend throughout their bodies and form an elaborate respiratory network [35, 36]. Although this unique respiratory system could provide a good target for pest control, there is still no established targeting method for insects.

For agents targeting the insect's respiratory system, tolfenpyrad and chlorfenapyr are types of chemical insecticides that inhibit the respiration of insects, which block not the tracheal development but the electron-transfer system of mitochondria [36, 37]. While their high insecticidal activity, since these insecticides target mitochondria that are common to most organisms, these insecticides are toxic to humans and other vertebrates as well and therefore considered to be with high environmental and health risks [37, 38]. Fumigants and oils extracted from plants are known to block the spiracle of insects, causing suffocation [39–41]. However, there is no method to prevent the trachea development.

Tracheas are rigid structures lined with a cuticle, which is replaced during each molt [42, 43]. Although transcription factor genes such as *ventral veinless* (*vvl*), *branchless* (*bnl*), and *trachealess* (*trh*) are known to be involved in the formation of tracheas [44, 45], the mechanisms responsible for their robustness have remained largely unknown. In our recent study investigating the gut symbiotic system of the soybean pest bug, *Riptortus pedestris* (S1 Fig), we discovered that dual oxidase (Duox), which is specifically expressed in the trachea, promotes dityrosine cross-linking of proteins lining them [46]. This process leads to tracheal stiffening and the formation and stabilization of the trachea network. Duox generates reactive oxygen species (ROS) via its C-terminal NADPH oxidase domain that produces $H_2O_2$ and its N-terminal peroxidase-homology domain that converts the $H_2O_2$ into hypochlorous acid [47, 48]. Notably, feeding *R. pedestris* an antioxidant cysteine derivative, *N*-acetyl-L-cysteine (NAC), dramatically inhibited tracheal formation [46]. Furthermore, some studies mentioned Duox has key roles for model insect Drosophila gut immunity and gut trachea integrity [49, 50]. These results suggest that pest control by targeting the insect's respiratory function is feasible.

Our previous study pointed out the potential use of antioxidant cysteine derivatives in controlling insect pests [46]. However, the following fundamental points are still under investigation: (1) which the derivatives, other than NAC, are effective against insects, (2) the degree of insecticidal activity associated with these derivatives, and (3) the extent to which the derivatives are effective against different insect species other than the bean bug *R. pedestris*. To address these questions, this study aims to screen for more effective antioxidants cysteine derivatives and demonstrate the possibility of using the derivatives in pest control technologies with low environmental risks for sustainable agriculture.

## Materials and methods

### Rearing experiments in the bean bug

The bean bug *R. pedestris* used in this study was originally collected from a soybean field in Tsukuba, Ibaraki, Japan, and maintained in the laboratory (S1 Fig). The bean bugs were reared with dried soybean seeds and distilled water containing 0.05% ascorbic acid in Petri dishes (90

mm diameter and 20 mm high) at 25°C under a long-day regimen (16 h light and 8 h dark). To establish symbiont-harboring insects, field-collected soil was supplied to the drinking water so that insects acquire the symbiont bacterium *Caballeronia insecticola* from the soil.

Both *R. pedestris* nymphs and adults were used in the experiments. For each treatment, ten insects were placed in plastic petri dishes and fed soybeans as food. The antioxidants were introduced by dripping them onto the cut cotton pad containing the insect's water supply. The mortality rate of the bean bugs was then measured over the course of 20 days. The treated insects were finally subjected to immunostaining for dityrosine networks to observe the tracheal development in the M4 region.

## Antioxidant compounds used in this study

The antioxidants used in this study were obtained from two companies (S2 Fig): L-Ascorbic acid, Red cabbage dye, Urea, *N*-Acetyl-L-cysteine (NAC), L-cysteine (L-Cys), L-cysteine methyl ester hydrochloride (L-CME), L-methionine (L-Met), L-methionine methyl ester hydrochloride (L-MME), and D-penicillamine (D-PA) were supplied from Fujifilm-Wako (Osaka Japan), and L-cysteine ethyl ester hydrochloride (L-CEE), D-cysteine hydrochloride (D-Cys), D-cysteine methyl ester hydrochloride (D-CME), and 2-amino ethanethiol (2-AET) were supplied from Tokyo Chemical Industry (Tokyo, Japan). The concentration of the antioxidants was adjusted to 10 mg/ml, and pH to 6.8 using NaOH (Fujifilm-Wako, Osaka, Japan) and measured with AS800 pH meter (AS ONE, Osaka, Japan). Solutions were replaced once every five days to maintain freshness. For the experiment of low-concentrate the derivatives, Ascorbic acid, NAC, L-Cys, L-CME, L-CEE, D-Cys, D-CME, D-PA, 2-AET, L-Met, and L-MME were adjusted to 1 mg/ml (pH 6.8). The control group was fed sterile water containing no derivatives.

## Immunostaining for dityrosine networks (DTN)

Dityrosine networks (DTN) were stained using immunostaining, as described previously [46]. First, water-provided bugs or the antioxidant-fed bugs were dissected in phosphate-buffered salts (PBS) pH 7.4 (Takara, Shiga, Japan), and the midguts were fixed with 4% paraformaldehyde (Fujifilm-Wako, Osaka, Japan) for 20 min at 25°C. The fixed midguts were then washed twice with PBS, and incubated with a blocking buffer of 1% BSA (Fujifilm-Wako, Osaka, Japan) in PBST (PBS containing 0.5% Tween20) (Fujifilm-Wako, Osaka, Japan) for 30 min at 25°C. The guts were subsequently incubated with 1:400 dilution of anti-dityrosine monoclonal antibody (MyBioScience, SAN, US) overnight at 4°C in a shaker (BioShaker BR-23FP; Taitec, Saitama, Japan). Following this step, the midguts were washed three times with PBS, each time for 10 min, and then stained with 1:500 dilution of goat anti-mouse immunoglobulin G, whose heavy and light chains conjugated with Alexa Fluor 555 (Abcam, Cambridge, UK), for 1 h at 25°C in the dark. Next, the guts were washed thrice with PBS, for 10 minutes each time. After washing, the gut samples were put in 200 μl of PBS and incubated with 1 μl of 300 μM 4′,6-diamidino-2-phenylindole (DAPI) (Thermo Fisher Scientific, MA, US), and 1 μl of 6 μM phalloidin Alexa-647 (Thermo Fisher Scientific, MA, US) for 1 hour in the dark. Finally, the midguts were transferred to a glass-bottom dish (Matsunami, Osaka, Japan), mounted with Prolong Gold Antifade Mountant (Thermo Fisher Scientific, MA, US), and covered by thick glass coverslips (Matsunami, Osaka, Japan). Samples stained this way were then observed with a laser-scanning confocal microscope (TCS SP8; Leica, Wetzlar, Germany). The DTN area (yellow parts in the fluorescent images) was measured using the software ImageJ [51].

## Various cysteine derivatives treatments for different insect species

Besides the bean bug *R. pedestris*, the following insect species were tested for sensitivity to the derivatives: green plant bug (*Apolygus spinolae*), red roach (*Blattea lateralis*), Japanese termite (*Reticulitermes speratus*), mealworm (*Tenbrio molitor*), and fruit fly (*Drosophila melanogaster*). *A. spinolae* were caught at a vineyard in Hokkaido, Japan. *B. lateralis*, *T. molitor* and *D. melanogaster* were purchased from a pet shop in Sapporo, Japan. *R. speratus* were caught on a forest path in Ibaraki, Japan. In all cases, each derivative was adjusted to a final concentration of 10 mg/ml and pH of 6.8. For the plant bug *A. spinolae*, ten adult insects were placed in a plastic petri dish (90 mm diameter and 20 mm high) and given plant seeds (soybeans, wheat, and sunflower) as a food source, water containing an antioxidant was also supplied on cut cotton pads. For the red roach, ten nymphs were reared in a plastic cup (97 mm diameter and 129 mm high) and fed tropical fish feed (Tetra, Melle, Germany); water containing the derivatives was supplied on cut cotton pads. For the termites, from 15 to 50 worker insects were placed in a plastic petri dish and given an 84 mm diameter filter paper (Advantec, Toyo Roshi, Japan) as their food. The filter papers were dropped with 3 ml of antioxidant solutions. For the mealworm, ten larvae were reared in a plastic petri dish and fed tropical fish feed which was mixed with the derivatives. For *Drosophila*, ten adult flies were placed in a plastic cup and fed a 10% honey solution wherein an antioxidant was mixed.

## Statistics

Statistical analysis was performed using Tukey's multiple comparison test (p < 0.05). Mann-Whitney *U* test was used to detect significant differences (p < 0.05) between the *R. pedestris* adult samples. Significant differences between the mortality rates in various insect species was determined using Fisher's exact test (p < 0.05). Statistical tests were conducted using Graph-Pad Prism v. 9.5.1 (Graph Pad Software, MA, US).

# Results

## Insecticidal activity of the antioxidants

Screening the antioxidants at 10 mg/ml concentration using *R. pedestris* nymphs showed that ascorbic acid, red cabbage dye, and urea had no insecticidal effect (S3 Fig). By contrast, chemical compounds with thiol or sulfide showed significantly high insecticidal activity (Fig 1A and 1B). Among them, 2-AET, L-Cys, L-CME, D-CME, and L-Met showed immediate effects, with all individuals dying within seven days (Fig 1B and S4 Fig). On the other hand, NAC, D-Cys, D-PA, and L-MME showed a slower effect (Fig 1B and S4 Fig). Particularly, NAC, D-PA, and L-MME tended to show high mortality after molting (S4 Fig).

## Trachea network formation after the feeding of the antioxidant cysteine and methionine derivatives

Although thick trachea and thinner ones (tracheoles) were both observed in *R. pedestris* crypt-bearing midgut region when DW or DW containing ascorbic acid or urea was fed (Fig 2A), the development of the tracheal network collapsed when nymphs were fed cysteine or methionine derivatives. Insects fed either NAC, L-Cys, L-MME, or 2-AET showed smaller tracheal networks in the midgut region (Fig 2B). Similarly, insects fed on L-Met also exhibited remarkably fewer numbers of tracheas (Fig 2C). Meanwhile, in insects fed on L-MME, some tracheae were observed, but their DTN signals seemed likely torn to pieces. The midgut tissue became smaller than those of control insects (Fig 2C). DTN areas of 4th instar fed cysteine and methionine derivative were smaller than DW, ascorbic acid and urea (Fig 2D and S1 Table).

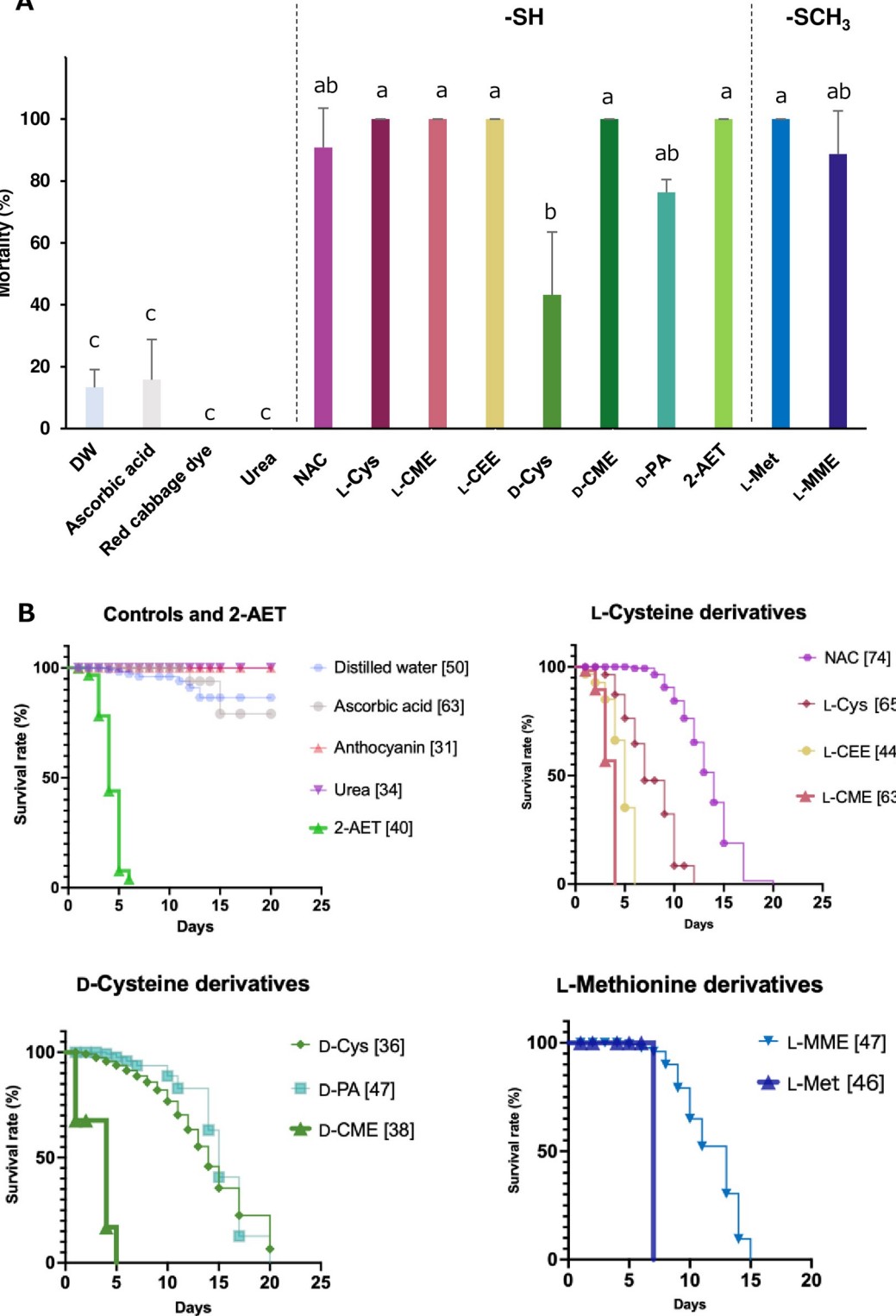

**Fig 1. Mortality and survival rate of antioxidants treated *R. pedestris*. (A)** Mortality of *R. pedestris* nymphs treated with each of the antioxidants (10 mg/ml, pH 6.8) after 14 days of treatment (mean ± SD, n = 3~5). In each replicate, 10 to 15 insects were reared in a petri dish, and mortality rate was measured. Observations started from the 3rd instar nymphs. Different letters indicate significant differences ($p < 0.05$, one-way ANOVA followed by Tukey's multiple comparisons). **(B)** Kaplan-Meier survival curves of R. pedestris nymphs treated with the antioxidants. Graphs are separated compound types.

The total numbers of insects at the starting time (day 0) are shown in brackets after the chemical names. Abbreviations: NAC, *N*-Acetyl-L-cysteine; L-Cys, L-cysteine; L-CME, L-cysteine methyl ester hydrochloride; L-CEE, L-cysteine ethyl ester hydrochloride; D-Cys, D-cysteine; D-CME, D-cysteine methyl ester hydrochloride; D-PA, D-penicillamine; 2-AET, 2-amino ethanethiol; L-Met, L-methionine; L-MME, L-methionine methyl ester hydrochloride.

## Insecticidal activity of the derivatives at a low concentration

To investigate the insecticidal activity of cysteine and methionine derivatives at a lower concentration, bean bug nymphs were fed the derivatives adjusted to 1 mg/ml. The result showed

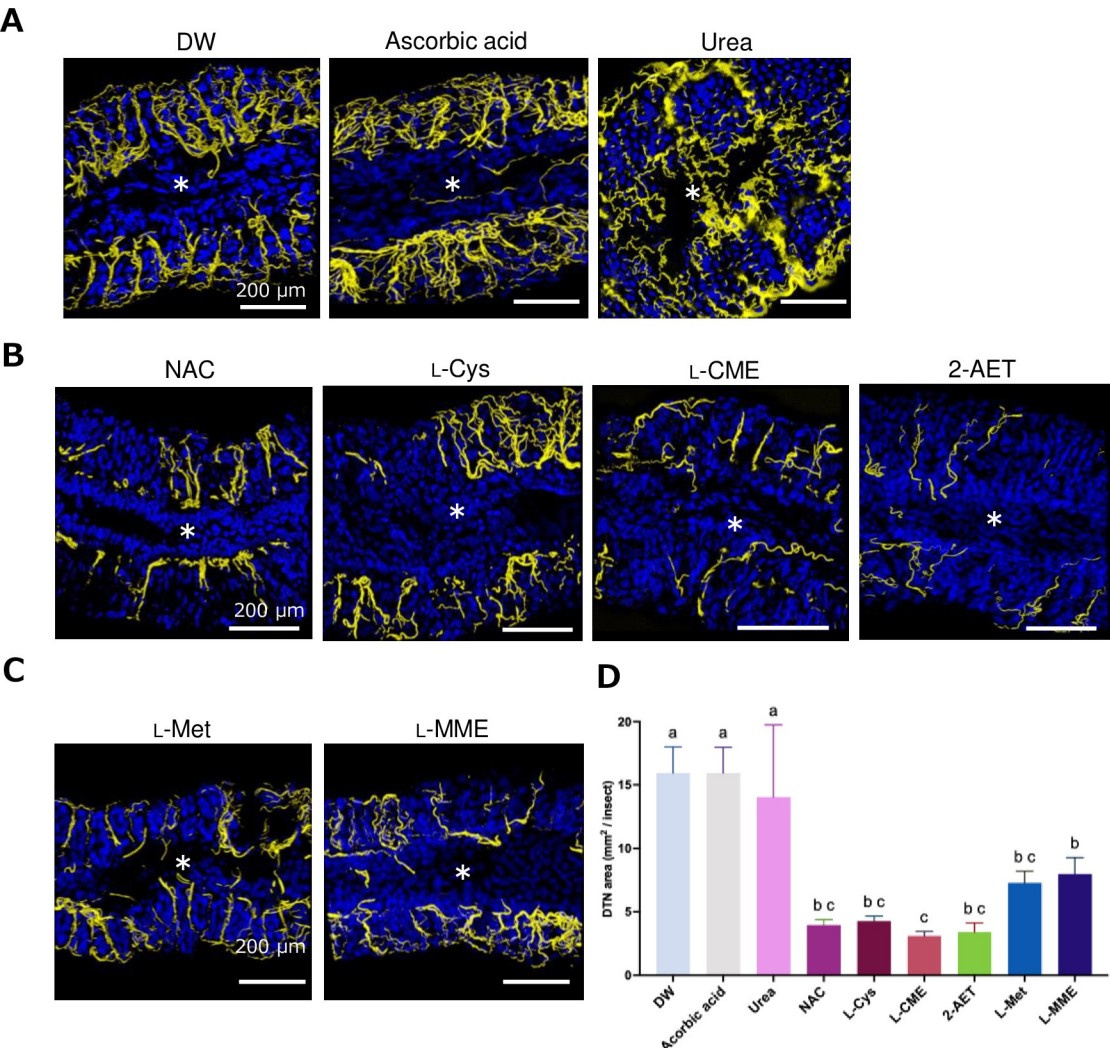

**Fig 2. Effects of the cysteine and methionine derivatives on the tracheal network development in the M4 gut region of *R. pedestris* 4<sup>th</sup> instar nymphs.** (**A**) Controls: distilled water (DW), ascorbic acid, and urea. (**B**) Sulfides: the methionine derivatives. (**C**) Thiols: the cysteine derivatives and 2-aminoethanthiol (2-AET). The constructed 3D pictures were taken 7 days after treatment and show the dityrosine network (DTN) and nuclei, as yellow and blue respectively. The asterisks indicate the main duct in the gut symbiotic organ of *R. pedestris*. (**D**) DTN area (mm²/insect). The [total DTN area/individual] was calculated from three pictures of each insect gut in which the DTN area (yellow parts in the fluorescent images) was measured using software ImageJ. The bar graph shows the mean ± SD of five individuals (n = 5). Different letters indicate significant differences ($p < 0.05$, one-way ANOVA followed by Tukey's multiple comparisons). Abbreviations: DW, Distilled water; NAC, *N*-Acetyl-L-cysteine; L-Cys, L-cysteine; L-CME, L-cysteine methyl ester hydrochloride; 2-AET, 2-amino ethanethiol; L-Met, L-methionine; L-MME, L-methionine methyl ester hydrochloride.

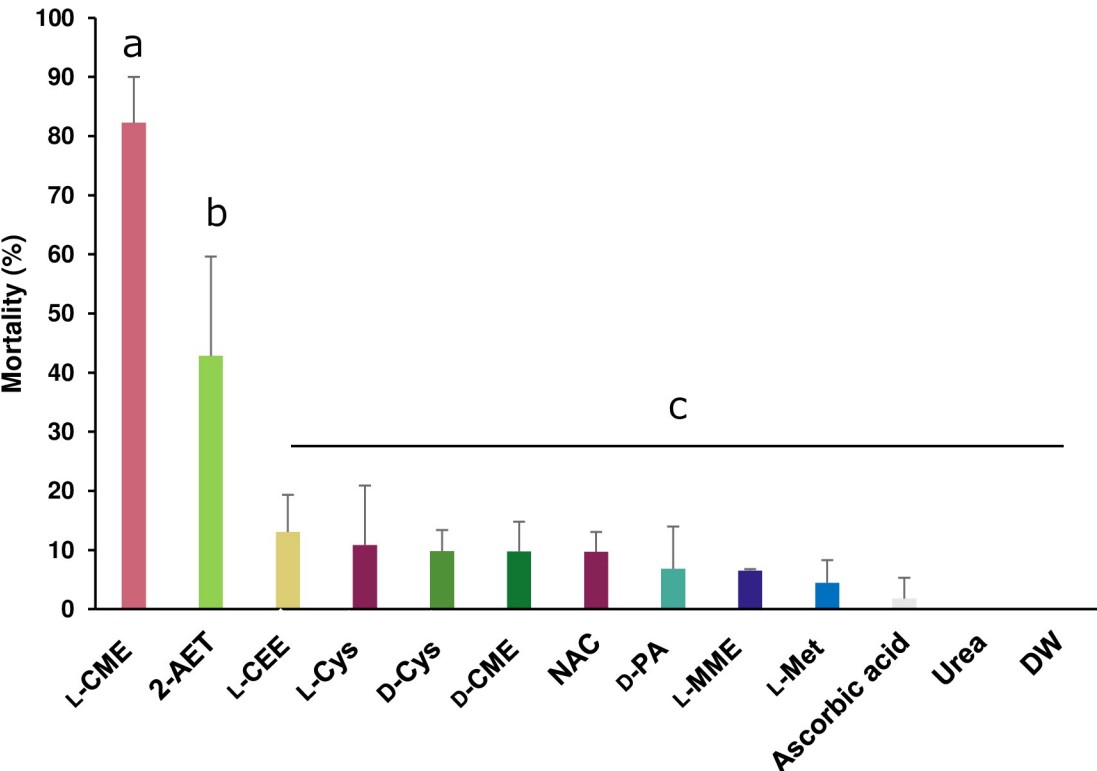

**Fig 3. Mortality of *R. pedestris* nymphs treated with the antioxidants adjusted to 1 mg/ml, pH 6.8.** Mortality (%) was measured 7 days after treatment (mean ± SD, n = 3). In each replicate, 10 to 15 insects were reared in a petri dish and mortality rate was measured. The test was conducted from $3^{rd}$ instar nymphs. Different letters indicate significant differences (p < 0.05, one-way ANOVA followed by Tukey's multiple comparisons). Abbreviations: DW, Distilled water; NAC, *N*-Acetyl-L-cysteine; L-Cys, L-cysteine; L-CME, L-cysteine methyl ester hydrochloride; L-CEE, L-cysteine ethyl ester hydrochloride; D-Cys, D-cysteine; D-CME, D-cysteine methyl ester hydrochloride; D-PA, D-penicillamine; 2-AET, 2-amino ethanethiol; L-Met, L-methionine; L-MME, L-methionine methyl ester hydrochloride.

that L-CME had the highest insecticidal activity, with more than 80% mortality observed seven days after starting treatment (Fig 3).

## Toxicity of the derivatives to adult bean bugs

Highly effective cysteine and methionine derivatives, including NAC, L-Cys, L-CME, 2-AET, L-Met, and L-MME, were selected and fed to adult bean bugs. Observations by immunostaining showed DTN development in adults treated with DW, ascorbic acid, and urea (Fig 4A). Meanwhile, tracheal DTN development was dramatically limited in adult insects treated with L-Met, L-MME, NAC, L-Cys, L-CME, or 2-AET (Fig 4B and 4C). The area DTN showed L-CME and 2-AET were smaller than others treatment (Fig 4D and S2 Table). In the first seven days of the treatment, L-CME exhibited the highest mortality (100%), followed by 2-AET (70%) (Fig 4E). The effects of the other tested derivatives were not significantly different from ascorbic acid (Fig 4E). There was no significant difference in insecticidal efficacy between male and female adults (S5 Fig).

## Insecticidal activity of antioxidant cysteine derivatives in different insect species

Based on the above results, the insecticidal activity of antioxidant cysteine derivatives was also investigated in the fruit fly (*D. melanogaster*), red roach (*B. lateralis*), Japanese termite (*R.

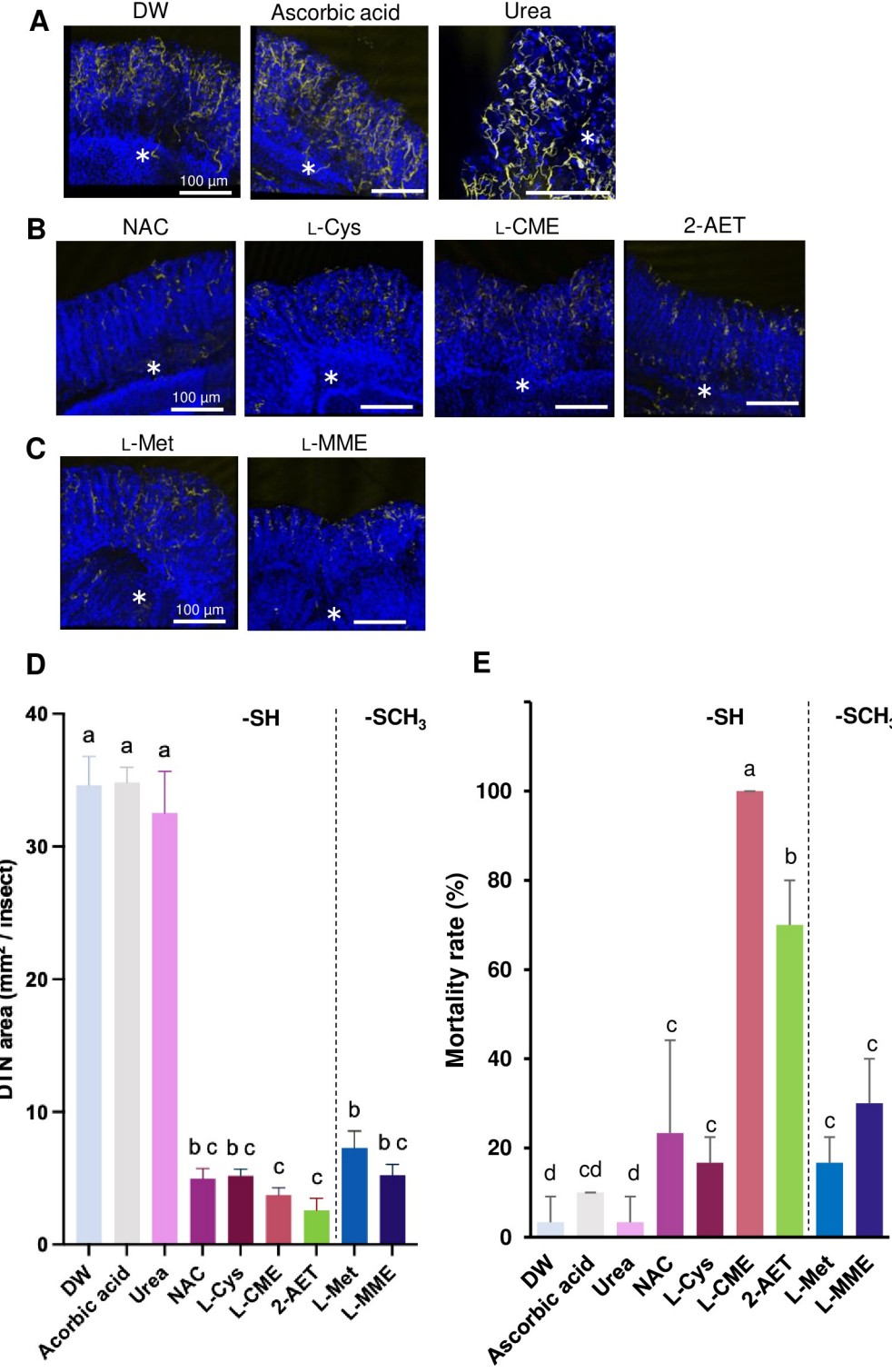

**Fig 4. Effects of the cysteine and methionine derivatives on adult *R. pedestris*.** (**A-C**) Effects of the derivatives on the tracheal network development in the M4 gut region of adult *R. pedestris*. Images were taken seven days after treatment. (**A**) Controls: distilled water, ascorbic acid, and urea. (**B**) Sulfides: the methionine derivatives. (**C**) Thiols: the cysteine derivatives and 2-amino ethanethiol. The pictures were reconstructed in 3D. The dityrosine network (DTN) and nuclei in the pictures were stained yellow and blue respectively. The asterisks indicate the main duct of the gut symbiotic organ. (**D**) DTN area (mm$^2$/insect). The [total of DTN area/individual] was calculated using five pictures of each insect gut wherein the DTN area (yellow parts in the fluorescent images) was measured using software ImageJ. The bar

graphs show the mean ± SD of five individuals (n = 5). (**E**) Mortality of adult *R. pedestris* treated with each of the derivatives (10 mg/ml, pH 6.8). The mortality was measured 7 days after treatment (mean ± SD, n = 3), and each replicated included ten individual insects. Different letters indicate significant differences (p < 0.05, one-way ANOVA followed by Tukey's multiple comparisons). Abbreviations: DW, Distilled water; NAC, *N*-Acetyl-L-cysteine; L-Cys, L-cysteine; L-CME, L-cysteine methyl ester hydrochloride; 2-AET, 2-amino ethanethiol; L-Met, L-methionine; L-MME, L-methionine methyl ester hydrochloride.

*speratus*), mealworm (*T. molitor*), and green plant bug (*A. spinolae*). The most effective derivatives differed among these insect species (Fig 5). L-Cys and L-CME were the most effective against the green plant bug *A. spinolae*, on the other hand, NAC was effective on the red roach

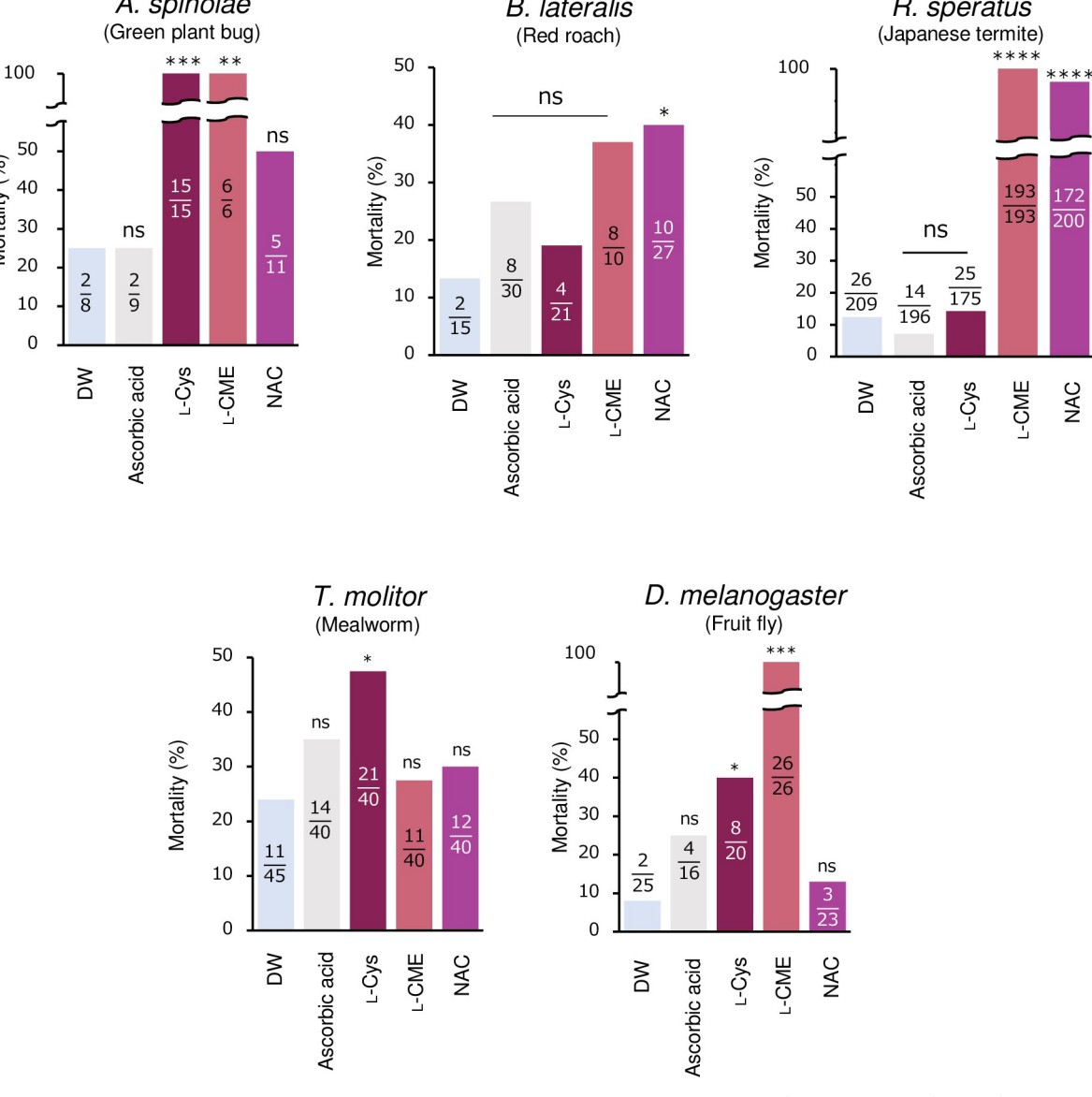

**Fig 5. Mortality of diverse insect species when they were fed the antioxidant cysteine derivatives (10 mg/ml, pH 6.8).** Mortality was observed at: 7 days after treatment in the green plant bug *A. spinolae*; 14 days after treatment in the red loach *B. lateralis* and the mealworm *T. molitor*; 2 days after treatment in *D. melanogaster*. The numbers of [dead insects/total tested insects] are shown in the bar graphs. Each treatment was compared with the distilled water treatment (*, p < 0.05; **, p < 0.01; ***, p < 0.005; ****, p < 0.001; Fisher's exact test). Abbreviations: DW, Distilled water; NAC, *N*-Acetyl-L-cysteine; L-Cys, L-cysteine; L-CME, L-cysteine methyl ester hydrochloride.

*B. lateralis* and *R. speratus*. L-Cys had the highest insecticidal activity against mealworms *T. molitor*, and L-CME displayed the highest insecticidal effect on fruit flies *D. melanogaster*.

## Discussion

Reactive oxygen species (ROS) are toxic substances that are always generated in aerobic organisms, including insects, when they use oxygen to carry out their vital activities [52]. Excessive production of ROS can damage cell membranes, accelerate aging, and negatively affect homeostasis in living organisms. At the same time, it is also known to be highly toxic to pathogens and other organisms that invade from the outside [53]. However, it has already been reported that in insects, ROS is also an important factor for DTN formation in many functional parts [54–57]. Compounds with sulfur atoms bonded to them were originally known as potent scavengers of these ROS [58–61]. Several research groups have already shown from human cell and animal experiments that L-Cys derivatives, a type of amino acid, function as effective antioxidant derivatives *in vivo* by exhibiting a strong ROS scavenging capacity [62, 63]. Our previous research indicated that NAC could effectively inhibit tracheal formation by suppressing ROS produced by Duox [46]; nevertheless, its insecticidal properties remained unclear until the present study, confirming that NAC has insecticidal activity (Fig 1A). In addition, our screening of antioxidant cysteine derivatives other than NAC demonstrated that several cysteine and methionine derivatives, including 2-AET, L-Cys, L-cysteine methyl ester (L-CME), D-cysteine methyl ester (D-CME), L-methionine (L-Met), D-cysteine (D-Cys), D-penicillamine (D-PA), and L-methionine methyl ester (L-MME) also exhibit insecticidal activity (Fig 1A and S4 Fig) by suppressing tracheal development (Fig 2). In contrast to these thiols (R-SH) and sulfide (R-S-R') compounds, no significant insecticidal effects were observed with the antioxidants such as ascorbic acid, urea, and red cabbage dye (S3 Fig), suggesting that thiols or sulfides can inhibit dityrosine formation in tracheal constituent proteins when they reach the trachea.

The derivatives 2-AET, L-cys, L-CME, D-CME, and L-Met showed relatively rapid insecticidal effects, leading to death within 7 days for all tested bean bugs (Figs 1B and 4). On the other hand, NAC, D-Cys, D-PA, and L-MME were slow acting (Figs 1B and 4). The variation in insecticidal activity among the derivatives is probably attributable to differences in their efficiency at reaching the trachea/tracheole, and their metabolic processes within insects. In a mammalian case, for instance, a recent study has revealed that L-Cys binds excess sulfur in the body and produces an active sulfur substance called "cysteine persulfide", which exhibits powerful ROS scavenging ability [59]. The high insecticidal potential of L-CME and L-Cys may be attributed to such a metabolic process. In this context, the variability in the inhibition level of tracheal network formation based on the compound used (Figs 2 and 4) may be caused by *in vivo* dynamics and the metabolic disposition of the different derivatives.

Among cysteine and methionine derivatives evaluated in this study, L-CME exhibited promising results due to its fast-acting properties and insecticidal efficiency. In addition, L-CME was effective not only at a concentration of 10 mg/ml but also at a low concentration of 1 mg/ml (Figs 1A and 3) showing its efficiency in small doses. While the physiological reason for the superior insecticidal activity of L-CME over the other tested compounds remains unclear, further investigation into the speed of tracheal corruption by using L-CME labeled with a stable isotope and the *in vivo* inhibitory activity of ROS may uncover more effective cysteine derivative for pest control.

We tested the antioxidant cysteine and methionine derivatives at high concentrations in this study. In general, cysteine and methionine, nonessential amino acids, are omnipresent in the environments and thought to be rapidly degraded by microorganisms and physical factors

in the environment, and no reports of negative effects of the derivatives in the environment on mammals have been identified [64]. In addition, L-Cys is approved as a flavoring agent by the FAO/WHO/Joint Expert Committee on Food Additives (JECFA) because exposure to L-Cys through food consumption is expected to be much greater than its use as a flavoring agent [65]. L-Cys is also considered "generally recognized as safe" as a flavor ingredient by the Flavor and Extract Manufacturers Association (FEMA) [66]. Therefore, even if cysteine is sprayed in high concentrations on agricultural fields, it would not have adverse effects on human health and low environmental risk.

In this study, the insecticidal effects of the antioxidant cysteine derivatives were investigated not only in the bean bug *R. pedestris* (Hemiptera) but in diverse groups of insects including the fruit fly *Drosophila melanogaster* (Diptera), mealworm *Tenebrio molitor* (Coleoptera), red roach *Blatta lateralis*, Japanese termite *Reticulitermes speratus* (Blattodea), and the green plant bug *Apolygus spinolae* (Hemiptera) (Fig 5). The results showed, interestingly, that the cysteine derivatives that showed insecticidal activity differed between insect species, probably due to differences in the dynamics of the chemical compounds in their bodies and the efficiency with which they reach the tracheal tissues as discussed above. In fact, different insect species have been reported to respond differently to the cysteine derivatives. For example, it has been reported that giving NAC to *D. melanogaster* slows age progression and increases lifespan [67]. Still, it has also been reported that giving NAC injections to cotton bollworm *Helicoverpa armigera* results in a shorter pupal period and lifespan [68]. The finding that each insect species showed slightly different effects (Fig 5) suggests that developing pest control technology could become more species-specific and less environmentally hazardous.

Since the trachea network formation is renewed during molting, it is expected that the collapse of the tracheal network would occur mainly in nymphal/larval stages but rarely in adulthood. In the bean bug, however, L-CME and 2-AET remarkably affected tracheal network formation at the adult stage (Fig 4D) and showed high insecticidal activity (Fig 4A), strongly suggesting that (1) the dityrosine network underlining tracheas is renewed after adult emergence and/or (2) such cysteine and methionine derivatives inhibit the development of dityrosine bonds in tracheoles, which are narrow chitin-less tubes coming off trachea branches and associated with each cell [49]. This could explain why the derivatives strongly influenced adult individuals of *Drosophila* and *A. spinolae* (Fig 5). Gas exchange in insects is determined almost entirely by the tracheal system, which is limited in its oxygen supply by size [69, 70]. Therefore, insects need to ensure the development of tracheas that match their body size in order to breathe adequately [71]. If tracheal development is inhibited, oxygen transport inside will also fail, resulting in a lack of oxygen needed for body movement. In the experiment of adult *R. pedestris*, DTN formation was inhibited even when treated with other the derivatives, such as NAC, L-Cys, L-Met, L-MME (Fig 4B). Although death did not occur in the experiment of the adult insects, we believe that this result suggests some effect on locomotion, reproduction, and other vital activities.

Various methods can be applied to these derivatives in agricultural fields but spraying of the compounds on crops as treatment appears to be one of the more realistic approaches, wherein biological effects of the cysteine derivatives on plants should be carefully investigated. Notably, NAC is known to inhibit bacterial biofilm formation due to disrupting disulfide bonds [72]. In fact, NAC has shown an antibacterial activity as biofilm formation inhibitor against notorious plant pathogens, such as *Ralstonia solanacearum* and *Xanthomonas citri* [73, 74]. Previous studies have reported that NAC, converted to glutathione in the plant body, can contribute to antioxidant and detoxification effects and enhance the plant's immune system and environmental tolerance [75, 76]. NAC and other derivatives are widely used as supplements for maintaining health and avoiding diseases related to oxidative stress, such as cancer,

atherosclerosis, and neurological diseases [77–81], and as medical drugs for Idiopathic Pulmonary Fibrosis [82–84]. Therefore, their safety for humans and other vertebrates is well-established in most antioxidant compounds. Moreover, NAC is effective with Blattodea insects in our experiment (Fig 5), so the derivative can be useful for not only agricultural pests but also sanitary pests in the future. Although the chemicals used in this study have not been reported to be toxic to humans or animals, their effects on crops must be clarified carefully before implementation and thus further research is needed to explore NAC and other cysteine derivatives as new pest control agents with low environmental impact and investigate their effects on crops and the soil.

## Supporting information

**S1 Fig. Pictures of *R. pedestris*.** (A) Female adult of *R. pedestris*. (B) Photograph of the whole midgut of *R. pedestris*. Abbreviations: M1, midgut first region; M2, midgut second region; M3, midgut third region; CR, constricted region; M4B, anterior bulb of midgut region; M4, midgut fourth region with crypts; H, hindgut. (C) Enlarged picture of the M4. (D) Developing trachea within the M4. The thin black tubes are trachea, observed here developing in the crypt. The asterisk represents the main duct passing through the middle of the M4.
(PPTX)

**S2 Fig. Chemical compounds used in this study.**
(PPTX)

**S3 Fig. The survival rate (%) of *R. pedestris* nymphs treated with distilled water, ascorbic acid, anthocyanin, and urea.** Survival rates and nymphal stages of *R. pedestris*. Observations started from the 3$^{rd}$ instar nymphs. The total numbers of insects at the starting time (day 0) are shown in brackets after the chemical names.
(PPTX)

**S4 Fig. Survival rates depend on the developmental stage of *R. pedestris*.** Observations started from the 3$^{rd}$ instar nymphs. Color indicates developmental stage: green, 3$^{rd}$ instar; blue, 4$^{th}$ instar; yellow, 5$^{th}$ instar; brown, adult. Red Arrows indicate when all individuals died. The total numbers of insects at the starting time (day 0) are shown in brackets after the chemical names. Abbreviations: NAC, *N*-Acetyl-L-cysteine; L-Cys, L-cysteine; L-CME, L-cysteine methyl ester hydrochloride; L-CEE, L-cysteine ethyl ester hydrochloride; D-Cys, D-cysteine; D-CME, D-cysteine methyl ester hydrochloride; D-PA, D-penicillamine; 2-AET, 2-amino ethanethiol; L-Met, L-methionine; L-MME, L-methionine methyl ester hydrochloride.
(PPTX)

**S5 Fig. Mortality (%) of *R. pedestris* adults fed the antioxidant cysteine and methionine derivatives.** Results of female (F) and male (M) are shown. Mortalities were observed 7 days after treatment (mean ± SD, n = 3). In each replicate, five female and five male individuals were reared in a petri dish, and their mortality was observed. Abbreviations: DW, Distilled water; NAC, *N*-Acetyl-L-cysteine; L-Cys, L-cysteine; L-CME, L-cysteine methyl ester hydrochloride; L-CEE, L-cysteine ethyl ester hydrochloride; D-Cys, D-cysteine; D-CME, D-cysteine methyl ester hydrochloride; D-PA, D-penicillamine; 2-AET, 2-amino ethanethiol; L-Met, L-methionine; L-MME, L-methionine methyl ester hydrochloride. The significant differences were determined by the Mann-Whitney *U* test (p < 0.05) between females and males. n.s.: not significant.
(PPTX)

**S1 Table. Details of DTN area in 4th instar of *R. pedestris*.**
(PDF)

**S2 Table. Details of DTN area in adults of *R. pedestris*.**
(PDF)

## Acknowledgments

We thank Madoka Miyazaki for rearing the insects. We thank Peter Mergaert, Hideomi Ito, Aya Yokota, and Antoine-Olivier Lirette for advising on this project.

## Author Contributions

**Conceptualization:** Yoshitomo Kikuchi.

**Data curation:** Hiroyuki Morimura, Shusei Kanie.

**Formal analysis:** Yuya Sato.

**Funding acquisition:** Yoshitomo Kikuchi.

**Investigation:** Hiroyuki Morimura, Kota Ishigami, Shusei Kanie, Yuya Sato, Yoshitomo Kikuchi.

**Project administration:** Yuya Sato, Yoshitomo Kikuchi.

**Supervision:** Yoshitomo Kikuchi.

**Writing – original draft:** Hiroyuki Morimura, Shusei Kanie, Yuya Sato.

**Writing – review & editing:** Yoshitomo Kikuchi.

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
