## [Decision Letter · Decision Letter 0]

18 Mar 2024

PONE-D-23-43528Antioxidant cysteine and methionine derivatives show trachea disruption in insectsPLOS ONE

Dear Dr. Kikuchi,

Thank you for submitting your manuscript to PLOS ONE. After careful consideration, we feel that it has merit but does not fully meet PLOS ONE’s publication criteria as it currently stands. Therefore, we invite you to submit a revised version of the manuscript that addresses the points raised during the review process.

First, I need to apologize for the delay, and explain to the authors that nine reviewers were invited and only one accepted. As I think a second opinion would help here, as your paper makes some claims that are not trivial, I chose to include my own comments  as a second reviewer in order not to cause further delays.

PLO (Academic Editor) comments :

The paper makes a relevant contribution, extending from previous work to include data from a broader set of compounds involved in thiol redox metabolism and evaluating the toxicity of these compounds against other species of insects. However, the claims on the use of these compounds as “general” insecticides seem to go against their own data and the literature mentioned by the authors, showing even lifespan-extending action in some cases (ref 66). The observation of different outcomes on the effects of distinct compounds and species may reflect that a more complex biology is involved than the authors proposed, which seems to limit the mechanism to DTN formation in trachea. Cuticle formation involves thiol redox and extensive protein cross-linking, and several molts occur between nymphal stages during the time interval analyzed. Thiol-based cell signaling is a major research subject in the redox biology field.

Note that 10 mg /ml (the concentration used in experiments) is about 0.1 M an extremely high concentration to be used as a spraying insecticide (insecticides are usually active at the nanomolar range). Also, insecticide action should be more appropriately evaluated as dose response curves, measuring the amount actually ingested/or applied to the insect. Therefore, although the claim that thiol redox metabolism may be used as a target in the development of new insecticides is still valid, I cannot see how, within this concentration range, these compounds could be used as insecticides. It seems that translational implications are still further ahead at this moment. Therefore, at least, these limitations should be discussed as a point that need to be addressed.

About reviewer #1 comments, I wuld like to acknowledge here that the reviewer who accepted the invitation has done a careful work and raised several points that must be addressed and can improve your manuscript. Particular attention should be given to comments (1) on the need to give replicate information missing in several figures. (2) Also, including DW controls that are mentioned but were not included in most figures is relevant. This implies that additional statistical tests against these DW control groups should be included. (3) Using ascorbic acid (a redox active molecule) as the single control does not appear reasonable.

We look forward to receiving your revised manuscript.

Kind regards,

Pedro L. Oliveira

Academic Editor

PLOS ONE

4. Thank you for uploading your study's underlying data set. Unfortunately, the repository you have noted in your Data Availability statement does not qualify as an acceptable data repository according to PLOS's standards.

5. We note that Figure 2, 4 and S1 in your submission contain copyrighted images. All PLOS content is published under the Creative Commons Attribution License (CC BY 4.0), which means that the manuscript, images, and Supporting Information files will be freely available online, and any third party is permitted to access, download, copy, distribute, and use these materials in any way, even commercially, with proper attribution. For more information, see our copyright guidelines: http://journals.plos.org/plosone/s/licenses-and-copyright.

1. You may seek permission from the original copyright holder of Figure 2, 4 and S1 to publish the content specifically under the CC BY 4.0 license. 

Reviewers' comments:

Reviewer's Responses to Questions

**Comments to the Author**

1. Is the manuscript technically sound, and do the data support the conclusions?

Reviewer #1: Partly

2. Has the statistical analysis been performed appropriately and rigorously? 

Reviewer #1: Yes

3. Have the authors made all data underlying the findings in their manuscript fully available?

Reviewer #1: Yes

4. Is the manuscript presented in an intelligible fashion and written in standard English?

Reviewer #1: No

5. Review Comments to the Author

Reviewer #1: This interesting manuscript by Morimura and colleagues identifies cysteine and methionine derivatives with antioxidant activity that inhibit insect tracheal formation and respiration and may potentially be used as insecticides. The work builds on a previous publication of the group where they found that in the hemipteran soybean pest bug, Riptortus pedestris, ROS production by Duox is key to the insect’s tracheal network stabilization, because ROS are necessary for di-tyrosine bond protein crosslinking. If Duox is silenced or ROS are inhibited by the antioxidant NAC (a cysteine derivative), insect respiration is blocked. In the current study, the authors screened a panel of antioxidant compounds (13 in total including NAC) for trachea deformation and insecticidal effects. They characterized the compounds for their speed of insecticidal action, their effects on the tracheal system of the soybean pest bug, Riptortus pedestris, and they also tested dosage requirements in Riptortus pedestris and the effects of some of the antioxidants (Cys-derivatives) on mortality of other insect species.

The authors need to fix the following issues before publication:

1. Page 2, Abstract, last sentence: if the compounds have broad insecticidal activity, as demonstrated for some of the Cys derivatives (e.g. L-CME, NAC), then they may not be good for pest control. Further dosage experiments should be done to conclude this. I suggest to edit the last sentence of the abstract to “Our results suggest that some antioxidant compounds have specific tracheal inhibitory activity in different insect species and they may be used as novel pest control agents upon further characterization”.

2. Page 4-5, lines 83-85 “Furthermore, some studies observed that … … [49, 50]”: this sentence needs rephrasing. In addition, the references provided (#49 and #50), although important for the Duox discussion, they seem irrelevant to the statement. Of note, the papers cited here have found key roles of Duox in gut immunity (#49) and gut trachea integrity (#50) in Drosophila.

3. Page 5, lines 93-95: the last sentence of the introduction needs rephrasing to depict the findings of the study. For example, it could start as “To address these questions, this study screened for more effective antioxidants with insecticidal action to demonstrate.….”.

4. Page 10, Fig. 1A: The water-feeding/negative control should be shown, irrespective of the minor effects of mortality by ascorbic acid, red cabbage dye and urea. How many independent replicates have been performed for this experiment? This information should be added in the figure legend in addition to the number of nymphs (10-15) used per replicate (which is stated).

5. Page 11, Fig. 2: the labels of the figure panels B and C are described in the reverse order they appear in the figure. The authors should either describe panel B (methionine derivatives) first and then panel C or reverse the two panels. (line 196: Fig. 2B refers to 2C; lines 197 and 199: Fig. 2C describes 2B).

6. Fig. 2: does ascorbic acid increase DTN signal density (this is the impression the reader gets from the images)? It would be best if the authors measured the tracheal network of each panel and also provided statistical significance of their results.

7. Fig. 3: the DW-fed control should be included.

8. Fig. 4A: the DW-fed control should be included; Fig. 4B-D: the authors should state what the asterisks indicate; tracheal network measurements would be nice; How do the authors explain that although L-CME and 2-AET are basically affecting adult survival, they exhibit the same phenotype (loss/reduction) in their DTN? Are these animals healthy? Did the authors try to follow adult survival at a later time point?

9. Fig. 5: how many times were these experiments performed? The authors should cross-check their statements in the methods and the numbers indicated on the graphs, because there are some inconsistencies.

10. Fig. S3: the DW controls should be included in the graphs. How many nymphs are used in these experiments? Are the experiments repeated?

11. Fig. S4: please see previous comment (#10).

12. The manuscript would benefit from English/text editing.

6. PLOS authors have the option to publish the peer review history of their article (what does this mean?). If published, this will include your full peer review and any attached files.

Reviewer #1: No

---

## [Author Response · Author response to Decision Letter 0]

20 Aug 2024

We appreciate the positive comments made by the reviewers regarding our manuscript entitled "Antioxidant cysteine and methionine derivatives show trachea disruption in insects” (PONE-D-23-43528). We have carefully revised our manuscript. You will find below a point-by-point reply to all comments (written in blue). Revised parts and sentences have been highlighted in the green marked-up manuscript.

PLO (Academic Editor) comments:

The paper makes a relevant contribution, extending from previous work to include data from a broader set of compounds involved in thiol redox metabolism and evaluating the toxicity of these compounds against other species of insects. However, the claims on the use of these compounds as “general” insecticides seem to go against their own data and the literature mentioned by the authors, showing even lifespan-extending action in some cases (ref 66). The observation of different outcomes on the effects of distinct compounds and species may reflect that a more complex biology is involved than the authors proposed, which seems to limit the mechanism to DTN formation in trachea. Cuticle formation involves thiol redox and extensive protein cross-linking, and several molts occur between nymphal stages during the time interval analyzed. Thiol-based cell signaling is a major research subject in the redox biology field.

Note that 10 mg /ml (the concentration used in experiments) is about 0.1 M an extremely high concentration to be used as a spraying insecticide (insecticides are usually active at the nanomolar range). Also, insecticide action should be more appropriately evaluated as dose response curves, measuring the amount actually ingested/or applied to the insect. Therefore, although the claim that thiol redox metabolism may be used as a target in the development of new insecticides is still valid, I cannot see how, within this concentration range, these compounds could be used as insecticides. It seems that translational implications are still further ahead at this moment. Therefore, at least, these limitations should be discussed as a point that need to be addressed.

>Thank you for your evaluation and constructive comments. Certainly, we agree that the effective concentration of antioxidants is high compared to those in common pesticides, therefore we have weakened the argument for their use as insecticides (e.g., Abstract [L31-33]). We have stated this study as a demonstration of its potential as a new pest control technology with low environmental load.

About reviewer #1 comments, I wuld like to acknowledge here that the reviewer who accepted the invitation has done a careful work and raised several points that must be addresssed and can improve your manuscript. Particular attention should be given to comments: 

(1) on the need to give replicate information missing in several figures. 

>The number of samples and repetitions are indicated in all figures (on panels and/or in footnotes) to avoid any confusion. 

(2) Also, including DW controls that are mentioned but were not included in most figures is relevant. This implies that additional statistical tests against these DW control groups should be included. 

>Thank you for the suggestion. We agree and have included DW as a control in this revised manuscript: Figure 1A, Figure 1B, Figure 3, Figure 4A, Figure S3 and Figure S5.

(3) Using ascorbic acid (a redox active molecule) as the single control does not appear reasonable.

> As an additional control, urea was tested in rearing experiments as long as possible. The results have been included in the revised manuscript: Figure 2A, Figure 3, Figure 4A, and Figure S5.

Reviewer 1: Review Comments to the Author

Reviewer #1: This interesting manuscript by Morimura and colleagues identifies cysteine and methionine derivatives with antioxidant activity that inhibit insect tracheal formation and respiration and may potentially be used as insecticides. The work builds on a previous publication of the group where they found that in the hemipteran soybean pest bug, Riptortus pedestris, ROS production by Duox is key to the insect’s tracheal network stabilization, because ROS are necessary for di-tyrosine bond protein crosslinking. If Duox is silenced or ROS are inhibited by the antioxidant NAC (a cysteine derivative), insect respiration is blocked. In the current study, the authors screened a panel of antioxidant compounds (13 in total including NAC) for trachea deformation and insecticidal effects. They characterized the compounds for their speed of insecticidal action, their effects on the tracheal system of the soybean pest bug, Riptortus pedestris, and they also tested dosage requirements in Riptortus pedestris and the effects of some of the antioxidants (Cys-derivatives) on mortality of other insect species.

>Thank you so much for your evaluation and useful comments on our manuscript.

The authors need to fix the following issues before publication:

1. Page 2, Abstract, last sentence: if the compounds have broad insecticidal activity, as demonstrated for some of the Cys derivatives (e.g. L-CME, NAC), then they may not be good for pest control. Further dosage experiments should be done to conclude this. I suggest to edit the last sentence of the abstract to “Our results suggest that some antioxidant compounds have specific tracheal inhibitory activity in different insect species and they may be used as novel pest control agents upon further characterization”.

>Thank you for the comment. The sentence has been revised as the reviewer suggested. (L31-33)

2. Page 4-5, lines 83-85 “Furthermore, some studies observed that … … [49, 50]”: this sentence needs rephrasing. In addition, the references provided (#49 and #50), although important for the Duox discussion, they seem irrelevant to the statement. Of note, the papers cited here have found key roles of Duox in gut immunity (#49) and gut trachea integrity (#50) in Drosophila.

>We agree, and the sentence has been revised to mention the different functions of Duox. (L83-84)

3. Page 5, lines 93-95: the last sentence of the introduction needs rephrasing to depict the findings of the study. For example, it could start as “To address these questions, this study screened for more effective antioxidants with insecticidal action to demonstrate.….”.

>The sentence has been revised according to the reviewer’s suggestion. (L92-94)

4. Page 10, Fig. 1A: The water-feeding/negative control should be shown, irrespective of the minor effects of mortality by ascorbic acid, red cabbage dye and urea. How many independent replicates have been performed for this experiment? This information should be added in the figure legend in addition to the number of nymphs (10-15) used per replicate (which is stated).

>For Figure 1A, the result of the DW-fed control has been added. 

For the experiment shown in the panel A, three replicates each of which consists of 10 to 15 insects were used. The bar graph shows means and standard deviations of the three replicates. This point is now noted in the footnote of Figure 1. 

In the panel B, The total numbers of insects at the starting time (day 0) are shown in brackets after the chemical names.

5. Page 11, Fig. 2: the labels of the figure panels B and C are described in the reverse order they appear in the figure. The authors should either describe panel B (methionine derivatives) first and then panel C or reverse the two panels. (line 196: Fig. 2B refers to 2C; lines 197 and 199: Fig. 2C describes 2B).

>Thank you. This mistake has been corrected. (L194-196; Figure 2B and C)

6. Fig. 2: does ascorbic acid increase DTN signal density (this is the impression the reader gets from the images)? It would be best if the authors measured the tracheal network of each panel and also provided statistical significance of their results.

>In this revised manuscript, we measured the DTN signal density by using ImageJ software to compare the tracheal development depending on the treatment. For measuring the DTN area, ImageJ software extracted only the yellow color representing DTN from each photo. The result is shown as Figure 2D. Based on this calculation, it was revealed that ascorbic acid does not significantly accelerate the trachea development compared with DW.

7. Fig. 3: the DW-fed control should be included.

>The DW-fed control has been added.

8. Fig. 4A: the DW-fed control should be included; Fig. 4B-D: the authors should state what the asterisks indicate; tracheal network measurements would be nice; How do the authors explain that although L-CME and 2-AET are basically affecting adult survival, they exhibit the same phenotype (loss/reduction) in their DTN? Are these animals healthy? Did the authors try to follow adult survival at a later time point?

>The DW-fed control has been included. 

Asterisks has been explained in the figure legend (Figure 4). This point was also lacking in Figure 2; now the description has been added in the figure legend. The tracheal development (DTN signals) has been measured by ImageJ, which is shown in Figure 2D, Figure 4E. As Figure 4D and E shows, insects with inhibited trachea development had higher mortality rates. Especially, L-CME and 2-AET treated adult insects that were seriously inhibited tracheal formation showed high mortality. 

The exact effect of antioxidants cannot be measured because the possibility of mortality due to a decrease in stamina and post-mating fitness must be affected once the insects reach the adult stage. For this reason, in this experiment with adult insects, mortality was measured in the short span of seven days after the antioxidants were treated. If measured after 14 days, mortality would be expected to increase for NAC and L-Cys. The correlation coefficient between DTN area and mortality in adults (measured after 7 days) was weak (R2 = 0.3752), but in the 4th instar (measured after 14 days) the correlation was very high (R2 = 0.886) (Please refer attached PDF file). Thus, although the observation period affected adult mortality, there is a correlation between DTN area and mortality, as indicated by the high correlation coefficient for 4th instar. Moreover, Jang et al. 2021 PNAS have indicated serious trachea collapsed R. pedestris with RNAi were almost died. These results suggest the tracheal development is involved in the survival rate of insects.

9. Fig. 5: how many times were these experiments performed? The authors should cross-check their statements in the methods and the numbers indicated on the graphs, because there are some inconsistencies.

> Because the number of individuals available for this experiment was different among insect species, the results of all individuals tested are merged and shown, different from Figures 1, 3-4, where laboratory-reared bean bugs were consistently used for the experiments. The mortality (dead insects/total tested) has been depicted at the top of each bar graph. Although the presentation style and statistical method differ from those of Figures 1, 3-4, there is no doubt that there are significant differences in effects among the antioxidants in different insect species. (Figure 5)

10. Fig. S3: the DW controls should be included in the graphs. How many nymphs are used in these experiments? Are the experiments repeated?

>The DW-fed control has been added. 

The panels shows the age change by time for all individuals used in Figure 1A. The number of insects is now described at the top of each figure. These points have been mentioned in the figure legend in this revised manuscript.

11. Fig. S4: please see previous comment (#10).

>As mentioned above (#10), each panel shows the instar change by time for all individuals used in Figure 1A. The number of insects is now described at the top of each figure. These points have been mentioned in the figure legend in this revised version.

12. The manuscript would benefit from English/text editing.

>This revised manuscript has been carefully checked by a native speaker.

---

## [Decision Letter · Decision Letter 1]

10 Sep 2024

Antioxidant cysteine and methionine derivatives show trachea disruption in insects

PONE-D-23-43528R1

Dear Dr. Kikuchi,

We’re pleased to inform you that your manuscript has been judged scientifically suitable for publication and will be formally accepted for publication once it meets all outstanding technical requirements.

Kind regards,

Pedro L. Oliveira

Academic Editor

PLOS ONE

Additional Editor Comments (optional):

Reviewers' comments:

Reviewer's Responses to Questions

**Comments to the Author**

1. If the authors have adequately addressed your comments raised in a previous round of review and you feel that this manuscript is now acceptable for publication, you may indicate that here to bypass the “Comments to the Author” section, enter your conflict of interest statement in the “Confidential to Editor” section, and submit your "Accept" recommendation.

Reviewer #1: All comments have been addressed

2. Is the manuscript technically sound, and do the data support the conclusions?

Reviewer #1: Yes

3. Has the statistical analysis been performed appropriately and rigorously? 

Reviewer #1: Yes

4. Have the authors made all data underlying the findings in their manuscript fully available?

Reviewer #1: Yes

5. Is the manuscript presented in an intelligible fashion and written in standard English?

Reviewer #1: Yes

6. Review Comments to the Author

Reviewer #1: The authors have provided a revised manuscript addressing all the questions raised in the first submission. I am happy with the revision of this interesting paper.

7. PLOS authors have the option to publish the peer review history of their article (what does this mean?). If published, this will include your full peer review and any attached files.

Reviewer #1: **Yes: **Chrysoula Pitsouli

---

## [Editor Report · Acceptance letter]

18 Oct 2024

PONE-D-23-43528R1 

PLOS ONE

Dear Dr. Kikuchi, 

I'm pleased to inform you that your manuscript has been deemed suitable for publication in PLOS ONE. Congratulations! Your manuscript is now being handed over to our production team.

Kind regards, 

on behalf of

Dr. Pedro L. Oliveira 

Academic Editor

PLOS ONE